# Electron Beam-Induced Compatibilization of PLA/PBAT Blends in Presence of Epoxidized Soybean Oil

**DOI:** 10.3390/polym15153265

**Published:** 2023-07-31

**Authors:** Lena Marbach, Philip Mörbitz

**Affiliations:** 1Department of Circular and Bio-Based Plastics, Fraunhofer UMSICHT, Institute for Environment, Safety and Energy Technology, Osterfelder Str. 3, 46047 Oberhausen, Germany; philip.moerbitz@umsicht.fraunhofer.de; 2Department of Chemistry and Biochemistry, Ruhr University Bochum, Universitaetsstr. 150, 44780 Bochum, Germany

**Keywords:** compatibilization, miscibility, PLA, PBAT, epoxidized soybean oil, electron beam treatment, biobased polymer

## Abstract

Blending of polymers can enhance performance of plastics and can give the opportunity to broaden the application fields. Especially the brittleness of poly(lactic acid) (PLA) is an issue, that is often addressed by blending it with soft polymers like poly(butylene adipate terephthalate) (PBAT). The immiscibility of those two polymers leads to limited properties of the blend. This study aimed to examine the application of electron-beam treatment with the implementation of a compatibilizing agent. PLA and PBAT were compounded with the addition of epoxidized soybean oil (ESBO) in different ratios and extruded into flat films. These were treated with electron beams at irradiation doses ranging from 12.5 to 100 kGy. The films thus produced were characterized by differential scanning calorimetry, size exclusion chromatography, scanning electron microscopy and tensile testing. A significant change in the glass transition temperatures of the blend partners was observed, as well as a substantial increase in elongation at break, even in PLA-rich compositions. These findings indicate improved compatibilization. Furthermore, the use of epoxidized soybean oil showed a changed extraction behavior of PBAT, indicating a formed binding to PLA. The results show that electron-beam treatment can significantly improve the compatibility between different polymers in blends, leading to enhanced mechanical and thermal properties.

## 1. Introduction

Due to the high consumption of plastics and the resulting accumulations in the environment, the use of bioplastics has increased. Poly(lactic acid) (PLA) is one biobased and biodegradable polymer, which has sufficient mechanical properties to replace fossil-based polymers in many applications. Nevertheless, its brittleness still limits its utilization. It is part of recent studies to blend PLA with other biobased and/or biodegradable polymers to enhance its elongation at break and impact strength. However, poor compatibility between different polymers in blends often limits their mechanical and physical properties [1,2]. 

The investigation of the miscibility of PLA with other polymers dates to the early 1990s. Initially, the emphasis was on blends with poly(caprolactone) (PCL), which were found to be considerably immiscible [3]. In subsequent research, this issue was addressed by utilizing reactive compatibilization with, e.g., triphenyl phosphite, which also led to significant enhancements in mechanical properties [4]. Nevertheless, reactive compatibilization remains a topic of ongoing research. Mörbitz et al. investigated the thermal and morphological properties of blends comprising poly(L-lactic acid)/poly(D-lactic acid)-B-poly(caprolactone) diblock copolymers. To create these copolymers, hydroxyl-terminated PCL oligomers were used as a macroinitiator for the ring-opening polymerization of D-lactide, resulting in a structure with plasticizing PCL units and compatibilizing poly(D-lactic acid) blocks. These block copolymers were then blended with a poly(L-lactic acid) matrix, leading to the formation of stereocomplex crystals. Compared to unbound poly(caprolactone), the PCL blocks exhibited a lower tendency to migrate in a solution test using toluene [5]. To overcome the brittleness of PLA, not only blends with PCL are used, but also numerous other ductile blend components. Here, too, the focus of the research projects is often on compatibility. Bai et al. investigated the compatibilization of PLA blends with poly(para-dioxane) by adding random copolymers thereof [6]. Also blends with fossil bases polymers like poly(ethylene) (PE) are investigated. Hillmyer et al. added diblock copolymers of PLA and PE to reach sufficient miscibility of the blends [7]. Reactive polymers are another sort of compatibilization technique. The reactive polymer should be miscible with one blend partner and reactive towards the other blend component. Blends of PLA and PBAT have been investigated to be compatibilized by reactive glycidyl methacrylate [8]. In general, there are many different blending systems with PLA, including natural rubber, thermoplastic starch or poly(ethylene terephthalate glycol) [9].

There are studies investigating the improvement of compatibility of PLA blends with poly(butylene adipate terephthalate) (PBAT) with epoxidized soybean oil (ESBO) as a renewable and environmentally friendly compatibilizer. Han et al. found that ESBO could undergo the epoxy ring-opening reaction with hydroxyl and carboxyl end groups of PLA and PBAT by melt blending, forming a network between the two polymers. The tensile strength could be increased from 36.2 to 43.3 MPa and elongation at break from 31 to 195%, respectively [10].

However, some polymers, like PLA, are easily degraded by thermal exposure, which is why Shin et al. investigated the compatibilization of PLA with PCL or starch in presence of glycidyl methacrylate using electron beams [11,12]. They also studied the influences on the rheological properties, impact strength and biodegradability of electron beam-irradiated PLA/PCL blends. The results show a significant improvement in interfacial adhesion and the storage modulus of all compatibilized blends was higher than that of uncompatibilized blends. The enhanced interfacial adhesion promotes morphological stability by acting as a network structure [13].

In general, electron-beam treatment of polymers is well-known for several applications such as crosslinked wires and cables or rotomolded drums. The electron beam-induced crosslinking of poly(ethylene) or poly(vinyl chloride) is often used to replace hazardous chemicals in the crosslinking process. Amongst this, the specific advantages of electron-beam treatment are, e.g., lower processing temperatures and separation of the shaping process of a product from the crosslinking step [14].

This study aims to combine the process of electron-beam treatment with the use of the biobased and environmentally friendly compatibilizer ESBO to use it on PLA/PBAT blends. So far, no studies have been carried out on the combination of PLA, PBAT, ESBO and electron irradiation. The blends should become reactively compatibilized to overcome the immiscibility. Blends of the two polymers are prepared with and without the additive and films are formed subsequently. The films are treated with electron beams at different irradiation doses, to examine the influence of the dose. To determine the compatibilization effect and overall properties of the blends DSC, SEC, SEM and tensile test were used. The novelty of this study is the application of electron-beam treatment in combination with renewable compatibilization agents on a PLA/PBAT system.

## 2. Miscibility of PLA and PBAT

For predicting the miscibility of PLA and PBAT, the solubility parameters can be calculated via a group contribution method. The solubility parameters after van Krevelen and Hoy are given in Table 1.

The mean solubility parameters can be used in the Flory–Huggins miscibility formula [16,17].
(1)$$\frac{∆G_{m}}{RTV}=\frac{\Phi_{a}}{V_{a}}ln\Phi_{a}+\frac{\Phi_{b}}{V_{b}}ln\Phi_{b}+\frac{\Phi_{a}\Phi_{b}}{RT}{\left(∆\delta \right)}^{2}$$

In this formula ΔG_m_ is the free enthalpy of mixing, V_a_ and V_b_ the molar volumes, Φ_a_ and Φ_b_ the volume fractions and Δδ the difference of the solubility parameters of the components. Applying this formula to systems with different molecular weights and different mass fractions of PBAT gives a miscibility diagram as shown in Figure 1. The designation 52.1/127 means a molar mass for PBAT of 52.1 kg/mol and 127 kg/mol for PLA. Since the densities of PLA and PBAT are very close, a density of 1.25 g/cm^3^ is assumed for both polymers.

It can be seen from the diagram that at higher temperatures the polymer fractions with molecular masses below 30 kg/mol for PBAT and below 50 kg/mol for PLA show miscibility and the system may be metastable at room temperature. However, systems that are miscible at elevated temperatures may become partially miscible or immiscible on cooling and may unmix over time [18]. It is therefore necessary to compatibilize the immiscible fractions to produce finely dispersed distributions or to stabilize the metastable mixed phases. Both can be achieved by forming block and graft copolymers at the phase interface.

## 3. Materials and Methods

### 3.1. Materials

Poly(lactic acid) (IngeoTM Biopolymer 2003D) was purchased from NatureWorks LLC, Minneapolis, MN, USA and poly(butylene adipate terephthalate) (ecoflex^®^ F Blend C1200) was obtained from BASF, Ludwigshafen, Germany. Epoxidized soybean oil (Merginat ESBO) was supplied from Hobum Oleochemicals, Hamburg, Germany. Tetrahydrofuran (THF), acetone and ethanol (Carl Roth GmbH & Co. KG, Karlsruhe, Germany) were used without further purification. Distilled hexafluoroisopropanol (HFIP, 99.9%) was purchased from ChemPur GmbH, Karlsruhe, Germany.

### 3.2. Preparation of PLA/PBAT Compounds and Films

The PLA and PBAT granulates were dried for 6 h at 60 °C. The compounds thereof were prepared using a double screw extruder (ZSK 25, Coperion GmbH, Stuttgart, Germany) at a mass ratio of 70/30, as already investigated by Han et al. [10]. The screw diameter was 25 mm with length-to-diameter ratio of 40. The processing temperature ranged from 60 °C to 170 °C throughout the different heating zones. The screw speed was 200 rpm. ESBO (7 phr) was dosed with a peristaltic pump (Thölen Pumpen GmbH, Geldern, Germany) into a mandrel of the extruder for one blend. A total of two blends called PLA+PBAT and PLA+PBAT+ESBO was produced. The produced granulates were dried again at 60 °C for 6 h and then formed into films with a thickness of 90 µm to 120 µm using a flat film extrusion machine (LCP-300, LabTech Engineering, Embourg, Belgium) with a single screw extruder. The temperature ranged from 190 °C to 215 °C. The die temperature was 180 °C. Screw speed was 70 rpm, the temperature of the chill roll was 25 °C and the haul-off speed was 3.9 m/min.

### 3.3. Electron-Beam Treatment

The produced film rolls were treated with electron beams during a roll-to-roll process using the electron beam unit atmoFlex 1250, Fraunhofer FEP, Dresden, Germany, as described by Günther et al. [19]. The acceleration voltage was 148 kV. The samples were treated with doses ranging from 12.5 kGy to 100 kGy under nitrogen atmosphere. The doses were found to be suitable by preliminary studies.

### 3.4. Differential Scanning Calorimetry (DSC)

Thermal analysis was carried out using a differential scanning calorimeter (DSC 204 F1 Phoenix, Netzsch, Selb, Germany). Glass transition temperatures and melting temperatures of the films were determined. The specimens underwent a two-step heating process. Initially, they were heated from 25 °C to 250 °C and subsequently cooled down to −50 °C. In the second step, the specimens were heated once more to 250 °C. Throughout both heating and cooling processes, a heating rate of 10 K/min was maintained under a nitrogen atmosphere. The focus of the investigation was on the second cycle.

### 3.5. Size Exclusion Chromatography (SEC)

Solutions of the films were prepared using distilled hexafluoro-2-propanol (HFIP) as solvent and a mixture of HFIP and acetone as internal standard. The flow rate of the solvent was 1 mL/min at a temperature of 23 °C. An Agilent 1100 instrument from Agilent, Santa Clara, USA was used for the SEC. Viscosity (ETA-2010, PSS GmbH, Mainz, Germany) and light-scattering (PSS SLD 7000, Brookhaven Instrument Inc, Holtsville, NY, USA) detectors were used for evaluation.

### 3.6. Scanning Electron Microscopy (SEM)

Before performing scanning electron microscopy, the films were soaked in THF for 48 h at room temperature to remove possible soluble fractions not bound to PLA. The objective of the electron-beam treatment was to facilitate the reactive binding of PBAT to PLA. If such binding had occurred, it should not be removed by THF during the soaking process. The treated films are then dried on a flat surface for 24 h at room temperature. SEM was carried out using a Vega 3 (TESCAN GmbH, Dortmund, Germany) with an acceleration voltage of 20 kV and a secondary electron detector. The samples were coated with 10 nm gold at a current of 40 mA using a sputter coater (Cressington 108auto, TESCAN GmbH, Dortmund, Germany).

### 3.7. Mechanical Properties

Tensile strength, elongation at tensile strength, elongation at break, Young’s modulus, breaking stress and tear propagation resistance were determined using a zwickiLine Z1.0 TH, Zwick Roell, Ulm, Germany. The tests were carried out using DIN EN ISO 527-3 using five specimens each. The samples were stored in a standardized climate (23 °C, 50% relative humidity) for 48 h beforehand.

## 4. Results and Discussion

### 4.1. Thermal Analysis

In Table 2 the thermal properties of the films are summarized. First, it can be stated that the blending of PLA and PBAT results in a lower glass transition temperature and melting temperature of PLA. The glass transition temperature of PBAT is increased by the blending, while the melting temperature seems to stay unchanged. In the absence of an additive, the melting point of PLA cannot be detected, as seen in Figure 2a, since the presence of PBAT is likely to inhibit crystal formation of PLA. Furthermore, electron treatment does not affect the melting behavior of PLA. However, when the additive is added, PLA exhibits two distinct and overlapping melting peaks, as can be seen in Figure 2b. The addition of ESBO appears to enhance chain mobility by accumulating as a small molecule between the polymer chains, leading to an increase in space between polymer chains. However, electron treatment counteracts this effect. One possible explanation is that ESBO undergoes a reaction with the polymers during treatment, becoming incorporated into the polymer network. Consequently, it can no longer enhance mobility as a small molecule between the chains. Moreover, electron treatment may have caused crosslinking between the polymers themselves and with each other, thereby preventing crystallization.

The melting enthalpy ΔH_m_ of PBAT also appears to be altered by irradiation. In the absence of the additive, irradiation at 25 kGy initially reduces the enthalpy from 2.7 J/g to 1.5 J/g and then increases it again to 2.3 J/g at 100 kGy. This could be since the branches formed by the irradiation hinder crystallization, so that the resulting melting enthalpy is also reduced. The effect is also visible for the additive. Here the melting enthalpy increases from 1.6 J/g at 25 kGy to 3.4 J/g at 100 kGy. In comparison to neat PBAT the melting enthalpy is rather low, which impairs the ability to crystallize. The melting enthalpy of PBAT at 0 kGy could not be determined as the recrystallization peak of PLA overlaps the melting peak of PBAT. It should be emphasized that the recrystallization peak only appears in the unirradiated sample with additive. This supports the hypothesis that the ESBO initially supports the mobility of the polymer chains.

Besides the effects on the crystallization ability, the electron treatment and the addition of ESBO also seem to have an influence on the glass transition temperatures. The glass temperatures converge by around 2 °C in the absence of an additive. With the addition of ESBO, the glass transition temperatures converge by almost 7 °C. The convergence of the glass transition temperatures may be an indication of the compatibilization of the two phases, as the glass transition temperature of a copolymer is made up of those of the individual polymers [20]. Similar findings were obtained from Heino et al., studying compatibilized poly(ethylene terephthalate)/poly(propylene) blends, where the compatibilization agent led to a shifted glass transition temperature of PET four degrees to that of neat PP [21].

### 4.2. Gel Permeation Chromatography

The electron-beam treatment results in changes in molecular weight and polydispersity index of the polymers, which were both measured with SEC. The results are given in Figure 3.

With increasing irradiation dose, the number average molecular weight decreases from around 80 to 50 kg/mol for the blend without additive. With ESBO added, the molecular weight decreases from around 80 to 60 kg/mol. The diagram also shows that the rate of decrease is attenuated with ESBO in the blend. It can be stated that the molecular weight decreases with increasing irradiation dose, which can be caused by two effects. One is that the electron-beam treatment leads to chain scission due to high energy input. The other effect is, as studied by Tobita et al., that the size of branched and crosslinked polymers in dilution is smaller than that of linear molecules. As the SEC is calibrated with linear poly(styrene) standards, it is very likely that the molecular weight is underestimated [22].

Furthermore, the polydispersity index was calculated. The diagram shows a decreasing PDI until an irradiation dose of 50 kGy and an increase at 100 kGy for both the blends without and with ESBO, probably due to chain scission at elevated irradiation doses. In general, the blend with ESBO added has slightly a lower PDI.

### 4.3. Scanning Electron Microscopy

SEM images were captured for both the untreated films and the films subjected to electron-beam treatment at doses of 50 kGy and 100 kGy, which can be seen in Figure 4.

The untreated films present a surface without any visible cavities. The samples without ESBO at 50 kGy and 100 kGy show cavities of a maximum of approximately 0.5 µm diameter, which may indicate extracted PBAT fragments. The sample treated with 100 kGy shows a larger number of cavities, implying more extracted PBAT fragments. The samples with ESBO added show no visible cavities, while the surface of the irradiated samples looks more homogeneous. The lack of cavities in the samples with ESBO suggests that the addition of the compatibilizing agent facilitated stronger interactions and binding between PBAT and PLA. This effect can be attributed to the presence of ESBO, which likely acted as a bridge between the two polymers, promoting their compatibility and reducing phase separation, resulting also in smaller PBAT phases.

Kilic et al. also used SEM to evaluate compatibilization of injection-molded PLA and PBAT blends with Epoxy-POSS nanoparticles. Uncompatibilized blends show a PBAT particle size of around 7 µm, while the additive leads to reduced PBAT droplet sizes of 3 µm. The smaller particle size can be traced back to improved interfacial adhesion and reduced surface tension between PLA and PBAT [23].

The cavity sizes determined in this study are much smaller than in comparable studies [1,24]. This is an indication that PBAT may not have been completely dissolved. To gain a more comprehensive understanding of the blend’s morphology and the extent of PBAT dissolution, cryo-fracture images of the samples could have provided valuable visual information. However, since films were used instead of injection-molded components, cryo-fractured surface images were not available in this study. To compensate for the lack of cryo-fracture images, dissolution tests were chosen. By this means, insights into the extent of compatibility and dispersion between the two polymers in the blend could be gained.

### 4.4. Mechanical Properties

Tensile tests were carried out on the films treated with different irradiation doses ranging between 12.5 kGy to 100 kGy. The nominal elongation at break, the Young’s modulus and the tensile strength are presented in Figure 5.

Without ESBO the nominal elongation at break shows no dependency on irradiation dose and stays in the range of 3%, representing the brittleness of PLA. The sole addition of ESBO shows a significant effect on the elongation at break. The unirradiated sample has an elongation at break of 268% and decreases to 172% at an irradiation dose of 12.5 kGy. A higher dose results in an increase up to a maximum of 338% at 37.5 kGy. An irradiation with high doses such as 100 kGy results in a drastic decrease until the magnitude of the initial value; this might be caused by chain scission processes.

Like the elongation at break, the Young’s modulus shows no significant dependency on the irradiation dose without the addition of ESBO. It shows high variations between 1840 and 2270 MPa, probably caused by phase separation of the two polymers and inhomogeneous distribution of the PBAT phases in the PLA matrix. The addition of ESBO initially results in a lower modulus of 1620 MPa, probably due to plasticizing effects. With increasing irradiation dose, the modulus increases to a range between 1820 and 1970 MPa, with a lower variation.

The addition of ESBO to the blend leads to a reduction of the tensile strength from 60 MPa to 50 MPa. This decrease suggests that the presence of ESBO influenced the mechanical properties of the blend, by acting as a plasticizer. Interestingly, without the additive, the electron-beam treatment alone showed no significant influence on the tensile strength. However, there was a slight decrease observed from 50 MPa to 44 MPa in the irradiated specimens with ESBO added, indicating a possible reaction of the additive.

In general, the error bars of the nominal elongation at break and the Young’s modulus display a notable variability, indicating possible heterogeneities of the polymer phases. Despite this, a tendency is visible and should be examined further. In context with the SEM images, it can be stated that the PBAT phases might be smaller through the compatibilizing effect of ESBO. The electron-beam treatment enhances this effect, probably caused by enhanced reaction between the epoxy groups of ESBO with the hydroxyl and carboxyl groups of the polymers, as also discussed by Han et al. [10].

## 5. Conclusions

The investigation into the compatibilization method for PLA/PBAT blends using electron-beam treatment in combination with epoxidized soybean oil (ESBO) yielded significant insights into the polymer blend’s properties.

One of the primary observations was the indication of compatibilization by the convergence of glass transition temperatures of PLA and PBAT through electron-beam treatment. This effect was further enhanced with the incorporation of ESBO into the blend. The electron treatment also influenced the molecular weight, leading to a decrease in chain length. However, the addition of ESBO helped mitigate this effect, resulting in a lower dispersity in the presence of the additive.

The scanning electron microscopy (SEM) images provided valuable information about the phase morphology of the blend. In samples without the additive, PBAT could be extracted, indicating a lack of compatibility between the two polymers. However, the presence of ESBO seemed to prevent this extraction, suggesting that PBAT might have been reactively bound to PLA, rendering it no longer soluble. Alternatively, the PBAT phases are smaller and more finely dispersed due to the compatibilizer and are therefore not visible in the images.

Moreover, the investigation revealed the influence of the additive on the elongation at break and Young’s modulus of the PLA/PBAT blend. Electron-beam treatment in combination with ESBO demonstrated an improvement in these properties, while treatment without the presence of the additive showed no significant influence.

The results show that the miscibility of PLA and PBAT can be improved by electron-beam treatment and the effects can be further enhanced by the combination with epoxidized soybean oil, as illustrated in Figure 6. Overall, this study demonstrated that electron-beam treatment, coupled with ESBO as a compatibilizing agent, is a promising approach to improve the properties and compatibility of PLA/PBAT blends. These findings offer valuable insights into the development of high-performance polymer blends with broadened application possibilities, contributing to the advancement of the plastics industry.

## Figures and Tables

**Figure 1 polymers-15-03265-f001:**
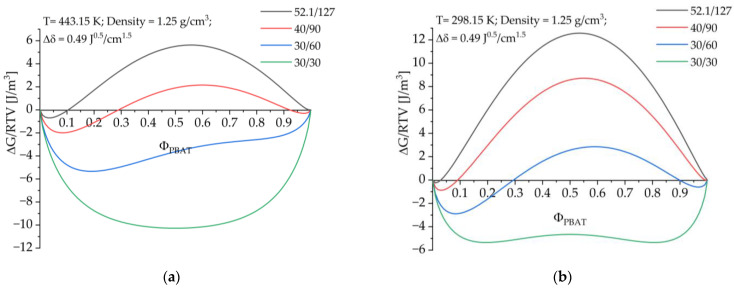
Prediction of the miscibility of PLA/PBAT blends with different weight ratios and molar masses at (**a**) 443.15 K and (**b**) 298.15 K.

**Figure 2 polymers-15-03265-f002:**
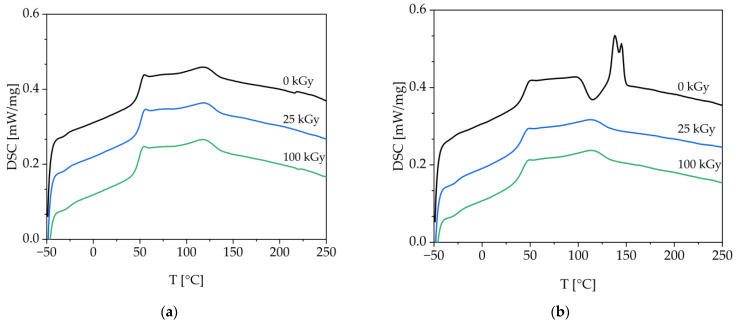
Thermograms of the PLA/PBAT without irradiation and with irradiation at 25 kGy and 100 kGy, (**a**) without ESBO, (**b**) with ESBO.

**Figure 3 polymers-15-03265-f003:**
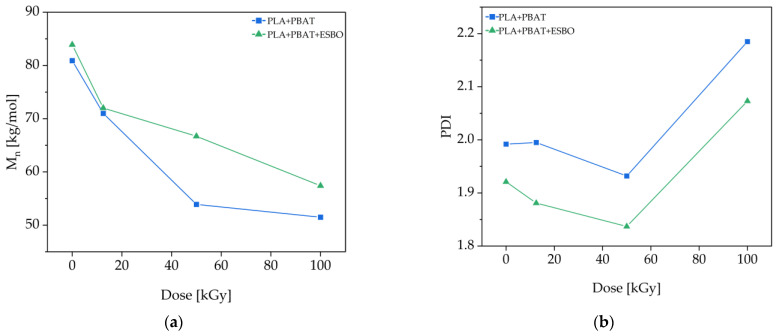
(**a**) Number average molecular weights (M_n_) and (**b**) polydispersity indices (PDI) of the films composed of PLA/PBAT and PLA/PBAT/ESBO.

**Figure 4 polymers-15-03265-f004:**
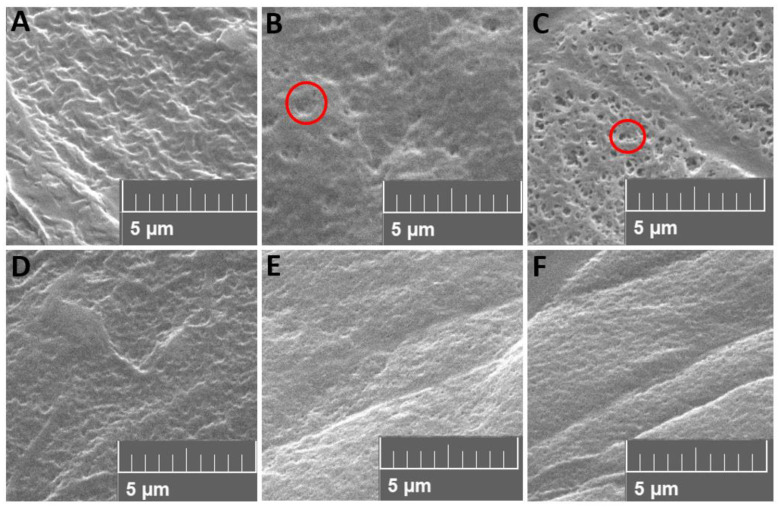
SEM of dried THF treated films with PLA/PBAT weight ratio of 70/30 at different doses: (**A**) 0 kGy without ESBO, (**B**) 50 kGy without ESBO, (**C**) 100 kGy without ESBO, (**D**) 0 kGy with 7% ESBO, (**E**) 50 kGy with 7% ESBO, (**F**) 100 kGy with 7% ESBO, the red circles indicate cavities.

**Figure 5 polymers-15-03265-f005:**
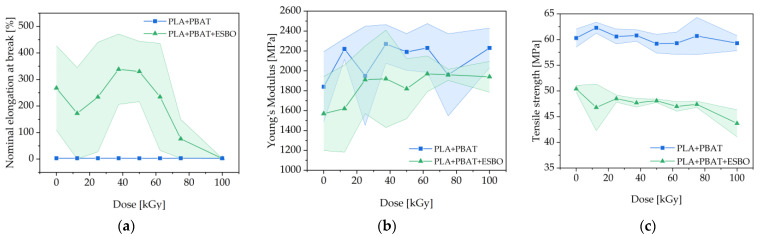
(**a**) Nominal elongation at break, (**b**) Young’s modulus and (**c**) Tensile strength of PLA/PBAT films without and with ESBO at different irradiation doses.

**Figure 6 polymers-15-03265-f006:**
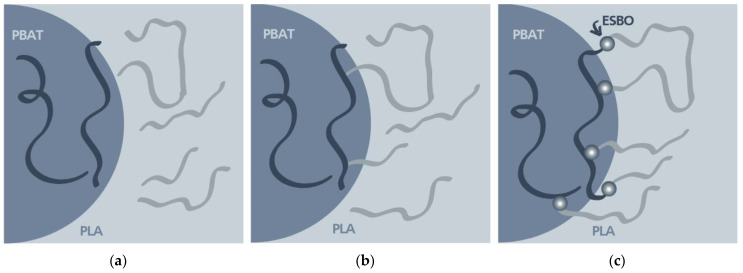
Schematic illustration of the proposed effects of the combination of electron-beam treatment with compatibilizing agents, (**a**) untreated blend, (**b**) blend after electron-beam treatment, (**c**) blend after electron-beam treatment under the presence of epoxidized soybean oil.

**Table 1 polymers-15-03265-t001:** Solubility parameters of PLA and PBAT [15].

Polymer Type	δ_van Krevelen_ [MPa^1/2^]	δ_Hoy_ [MPa^1/2^]	δ_Ø_ [MPa^1/2^]
PLA	20.66	21.31	20.99
PBAT	21.22	21.73	21.48

**Table 2 polymers-15-03265-t002:** Thermal properties of neat PLA and PBAT and of PLA/PBAT blends (a) without ESBO and (b) with the addition of ESBO, treated with different irradiation doses.

Composition	Dose [kGy]	T_m PLA_ [°C]	T_m PBAT_ [°C]	ΔH_m PLA_ [J/g]	ΔH_m PBAT_ [J/g]	T_g PLA_ [°C]	T_g PBAT_ [°C]
PLA	0	149.6	-	0.42	-	61.0	-
PBAT	0	-	120.4	-	14.9	-	−27.3
PLA/PBAT (a)	0	-	120.0	-	2.7	50.1	−26.9
	25	-	121.1	-	1.5	51.3	−28.3
	100	-	120.1	-	2.3	49.5	−25.8
PLA/PBAT/ESBO (b)	0	138.4 145.3	-	6.43 2.59	-	44.7	−28.1
	25	-	114.6	-	1.6	43.0	−26.3
	100	-	113.0	-	3.4	41.3	−24.7

## Data Availability

Not applicable.

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
