# Peer review of "Electron Beam-Induced Compatibilization of PLA/PBAT Blends in Presence of Epoxidized Soybean Oil"

_polymers, 2023, doi:10.3390/polym15153265_

Round 1

Reviewer 1 Report

Reviewer Comments

Abstract

1.       Line 21-22 – Changed extraction behavior refers to what properties in these blends/films? Please emphasize, is the characteristic or term of extraction.

Introduction

1.       2nd Paragraph, Line 37-50 – In this part, the author should focus on the blending process that involves a lot of polymers, not only focus on PCL. The process of the reaction was detailed in this paragraph. Other types of polymers could blend with PLA thus enhancing the properties. Please update this paragraph with the blending of PLA with other polymers.

2.       3rd Paragraph, Line 51-56 – This paragraph should include the application of the electron beam irradiated with polymers. The author should elaborate on the use of electron beam in this paragraph after the application of ESBO as a compatibilizer. This literature will give a clear picture of the research and what is all about.

3.       Last paragraph – The author should write the gap between the studies and the objectives of this research paper. The author also could assume the outcome of this research based on the objectives and the properties that have been investigated.

Miscibility of PLA and PBAT

1.       This subchapter could also include the mechanism of the electron beam treatment.

4.       The last paragraph of subchapter 2. / Line 93-101 could be the last paragraph for the introduction.

Materials and Method

1.       Line 114 – The author should elaborate a little bit about the ratio here. Why author decide to use the ratio? Is it any preliminary or justification?

2.       The preparation of the electron beam treatment could be in the diagram to show the mechanism. The author could consider drawing a diagram for this method.

3.       Line 130-131 – Why author choose these doses ranging? Any preliminary study or justification?

Results and Discussion

1.       In Figure 2 (b), the significant changes of thermograms 0 kGy not discuss further. The author needs to highlight and relate to the present discussion.

2.       In Figure 4, the significant difference should be remarked on by the author in the SEM micrograph.

3.       In Mechanical properties, the tensile should have 3 properties, Strength, Modulus and elongation. The author didn’t present the strength of the films.

Conclusion

1.       In Figure 6, the author should remark on the agents.

Overall comments

Overall, the paper is well-written and the research is sound. The introduction clearly states the purpose of the study, the methods section is well-organized and easy to follow, and the results are presented clearly and concisely.

Reviewer 2 Report

Authors investigated the selected properties of polymeric blends composed two commercially available polymers, of poly(lactic acid) (PLA) and poly(butylene adipate-co-terephthalate) (PBAT). The obtained blends were prepared with the use of compatibilizing agent, epoxidized soybean oil (ESBO) and treated with electron beams at different irradation doses. The aim of presented study is well stated, ESBO was not used in combination with electron beams irradiation, by now. The article is very interesting and the methods of material characterization are well chosen, however, comment or  explanation is required about following questions.

1.     Materials and Methods.

A.    Line 114; The composition were 70/30 PLA/PBAT and 65/28/7 …. Is it weight ratio?

B.    Line 133; DSC. In general more details of measurements should be given, such as full range of temperature during first and second heating run and cooling run (if it was done), nitrogen atmosphere?

C.    Line 140; SEC. A full name of solvent HFIP should be written. Please write also temperature and flow rate of the solvent during measurements.

D.    Line 146; SEM. What was a reason for removing possible soluble fractions from the blends by use of THF before SEM observations?

2.     Results and Discussion.

A.    Line 162; Thermal analysis. In my opinion, DSC measurements should be carried out also for pure components of the blends, i.e. PLA and PBAT separately. Primarily, for comparison with the results obtained for blends. All results should be given in Table 2.  

Line 178; Thermograms of blends. Glass transition temperature of PBAT in blends is below 0 °C, why it is not visible in presented thermograms? How Tg of PBAT was determined from thermograms? It can be suppose that ESBO improves miscibility of PLA and PBAT because their Tgs are closer to each other, as it can be seen from data in the Table 2, but it is not visible in thermograms.  

B.    Line 224; SEM. In my opinion, it can not be seen the existence of phase separations at presented SEM images. Why the dissolution tests was chosen? The disspersed phase (PBAT) in PLA matrix should be visible without dissolution. Please explain this.

The transmittance electron microscopic (TEM) could be helpful to characterize the morphology of prepared blends.

C.    Line 249; Mechanical properties. How many samples of each blend were used in measurements? Figure 5, please put the error bars for presented values of elongation at break and Young’s modulus.

 3.     Conclusions.

Please, reconsider the conclusion that electron beam treatment improves the miscibility of PLA and PBAT. Differences of Tgs values between untreated and irradiated blends are insignificant, almost the same.

In my opinion English Language is rather correct.

Round 2

Reviewer 2 Report

In my opinion the manuscript has been supplemented and corrected to a sufficient extent. In its current version can be published.